# Pharmacists’ Knowledge, Attitude and Practice Regarding the Dispensing of Antibiotics without Prescription in Tanzania: An Explorative Cross-Sectional Study

**DOI:** 10.3390/pharmacy8040238

**Published:** 2020-12-13

**Authors:** Baraka P. Poyongo, Raphael Zozimus Sangeda

**Affiliations:** Department of Pharmaceutical Microbiology, Muhimbili University of Health and Allied Sciences, Dar es Salaam P.O. Box 65013, Tanzania; barakapoyongo@gmail.com

**Keywords:** antibiotics, antibiotics resistance, antimicrobial resistance, dispensing, pharmacist, prescription, Tanzania

## Abstract

Inappropriate use of antibiotics has been reported to contribute to the emergence and increase of antimicrobial resistance (AMR) in the world. The pharmacist has the responsibility to supervise the dispensing of antibiotics with prescriptions to ensure rational use. An online semi-structured questionnaire was shared with approximately 1100 licensed pharmacists in Tanzania. Study data were collected and managed using REDCap electronic data capture tools before use for analysis. Of the 226 (20.5%) received responses, 197 had given consent and provided complete surveys. Notably, 153 (77.7%) of the 197 pharmacists had excellent knowledge about the legal requirements for dispensing antibiotics and the AMR challenge. Of the 197 surveyed pharmacists, 143 (72.6%) admitted to dispensing antibiotics without a prescription in their daily practice. Notably, 84.1% (37/44) of pharmacists with masters or PhD education were more likely to dispense without a prescription compared to 69.3% (106/153) among bachelor holders (*p*-value = 0.04). The reasons for administering antibiotics without a prescription included the pharmacy business looking for more profit, patient failure to obtain a prescription and the lack of stringent inspection of pharmacies by the regulatory authorities. Penicillins, macrolides and fluoroquinolones were the classes of antibiotics most commonly dispensed without a prescription. Stringent inspections by the regulatory authorities should detect and reduce dispensing antibiotics without a prescription. The community should be educated on the importance of medication prescription from a qualified medical practitioner.

## 1. Introduction

Antimicrobial resistance (AMR) is a growing problem in healthcare systems and occurs when previously susceptible microorganisms develop resistance to the previously effective antimicrobials [1]. Both appropriate and inappropriate use of antimicrobial agents has led to the development of AMR. Besides, some bacteria may have intrinsic structural and functional characteristics that make them naturally resistant to antibiotics [2]. However, inappropriate use of antimicrobials such as underuse, overuse or misuse are the leading causes for the development of AMR yet are preventable [3]. AMR has many consequences to both individual patients and the public at large, such as increasing cost, the burden of treatment, poor treatment outcome, and spread of the resistant microbes to other people hence endangering the public health [4,5].

AMR poses a significant threat to clinical efficacy, especially in the low-middle income countries (LMICs), because they lack comprehensive measures such as antimicrobial stewardship programmes (ASPs) and National Action Plans (NAP) to combat AMR [6,7,8]. There are growing concerns that pharmacists and other drug dispensers do not adhere to the Good Dispensing Practices (GDP), especially in lower and middle-income countries and thus contributing to AMR development [9].

Licensed pharmacists in community pharmacies have responsibility for dispensing antibiotics [10] in a rational manner. In many countries globally, including Tanzania, antibiotics are prescription-only medicine meant to be dispensed only to a person with a prescription from an authorized physician; this promotes the appropriate use of antibiotics hence reducing the rate of AMR [11]. Dispensing of antibiotics without a prescription in the community pharmacies is one of the contributing factors for the inappropriate use of antibiotics, thus enhancing the emergence and spread of AMR [12,13]. World Health Organization (WHO) urges that the fight against AMR should involve collaboration among the national experts to understand all aspects of antibiotic access, use and resistance within their country context. These experts then work on drafting and implementing policy solutions tailored to meet the country’s needs [14].

Retail drug shops are a primary source of care by many consumers in LMICs, especially those in rural or peri-urban areas lacking easy access to full-service of the public pharmacies. In 2003, Tanzania launched the accredited drug dispensing outlet (ADDO) program aiming at improving the quality of pharmaceuticals products and services at Part II retail medicine shops [15]. These part II drug stores, also called duka la dawa baridi (DLDB) (translated as essential medicine drug outlets) used to provide the selected non-prescription medicines. The problem with DLDB was illegal sales of prescription-only medicines [16]. On the one hand, the regulation of private pharmacy practice in Tanzania is regulated by the Pharmacy Council (PC) of Tanzania. On the other hand, regulation of quality, safety and effectiveness of medicines, medical devices and diagnostics is the responsibility of the Tanzania Medicines and Medical Devices Authority (TMDA).

A few surveys have been conducted in Tanzania to evaluate dispensing practices of antibiotics in community pharmacies [17,18,19]. These studies investigated whether medicine dispensers in community pharmacies or ADDOs dispense without a prescription regardless of their cadre, which could be a licensed pharmacist, pharmaceutical technician or pharmaceutical assistant. Since pharmacists are entitled to supervise the dispensing practice in pharmacies, their knowledge, attitude and practice are of concern as it may contribute to the inappropriate use of antibiotics and AMR development.

However, there are insufficient studies to evaluate pharmacists’ knowledge, attitude and practice regarding the dispensing of antibiotics without a prescription in Tanzania. Therefore, this study aimed to determine the proportion of pharmacists who dispense without a prescription, reasons for dispensing antibiotics without a prescription and their awareness of the legal requirements to dispense antibiotics and AMR.

## 2. Materials and Methods

### 2.1. Study Site

The study was conducted in all regions of Tanzania, a country found in East Africa. In 2019, Tanzania was estimated to have a population of 55 million inhabitants (approximation based on the 2012 national census) [20] and a total of 1863 licensed pharmacists [21].

### 2.2. Study Design and Sampling

This was a descriptive cross-sectional study of licensed pharmacists in Tanzania. The study was conducted from January 2019 to July 2019. The sample size was calculated using the single population formula based on previous research with a prevalence (P) of dispensing antibiotics without a prescription of 90 [17], considering 95% confidence interval (CI) and 0.05% margin of error given the formula
*n* = Z^2^P(1−P)/d^2^(1)
where *n* is a sample size, Z  =  1.96 for 95% confidence level, d = 0.05 is the marginal error, resulting in a sample size of 138. However, we used all the 197 responses we received from pharmacists who completed the questionnaire. This sample size had the power of 83% to detect the difference in attitude between higher education levels compared to undergraduate level education among pharmacists.

### 2.3. Study Tools

An online semi-structured questionnaire (Appendix A) with both open and mostly closed-ended questions was designed after reviewing and adopting tools from recent studies investigating dispensing antibiotics without a prescription [2,6,10,11] for use in this study. The tools were piloted and pretested by sending to 10 pharmacists and afterwards adjusting the language following the responses and feedback given. An invitation link was shared with pharmacists through WhatsApp groups where most of the licensed pharmacists participate. Besides, a list of names, contacts and emails of licensed pharmacists obtained from the Pharmacy Council of Tanzania were used to contact and request pharmacists to fill the questionnaire directly. Approximately 1100 licensed pharmacists across the country were invited to participate in this manner. A REDCap (Research Electronic Data Capture) survey was created for online data collection of pharmacists’ responses.

Data were collected and managed using REDCap; electronic data capture tools hosted at Muhimbili University of Health and Allied Science. REDCap is a secure, web-based application designed to support data capture for research studies, providing (1) an intuitive interface for validated data entry; (2) audit trails for tracking data manipulation and export procedures; (3) automated export procedures for seamless data downloads to standard statistical packages; and (4) procedures for importing data from external sources [22,23].

The questionnaire had five sections; the first section was for obtaining social-demographic characteristics of respondents such as gender, age, level of education, country of graduation of bachelor of pharmacy, job status in community pharmacy, region of practice and experience. The second section aimed to evaluate pharmacists’ knowledge of legal needs to dispense antibiotics and to assess pharmacists’ attitudes towards the dispensing of antibiotics without a prescription. Questions in this section investigated if pharmacists were aware that the dispensing of antibiotics without a prescription is an illegal practice in Tanzania which can lead to the emergence and spread of AMR. Furthermore, they were probed on problems that may arise upon dispensing antibiotics without a prescription and whether pharmacists should stop this practice.

The third section of the questionnaire explored the reasons that influence the dispensing of antibiotics without a prescription. These reasons were divided into two categories, which are patients based and pharmacists based reasons. Patient-based factors that drive a patient to visit a pharmacy without a prescription when they are sick were investigated. Such factors may be the reluctance of patients to spend many hours of waiting in a hospital, lack of insurance cover, the unwillingness of patients to see the doctor unless the disease is severe and patients’ inability to afford the cost for consultation and laboratory tests. Other reasons were provided by the pharmacists selecting a factor that mostly influence patients to visit pharmacy without a prescription. We evaluated pharmacists based reasons for dispensing antibiotics without a prescription, including the profit-oriented nature of pharmacy business, the relationship between the pharmacist and the patient and the belief that pharmacists are knowledgeable enough to dispense without a prescription. Additionally, we assessed the influence of regular inspection of pharmacies by the regulatory authorities on dispensing antibiotics without a prescription.

The fourth section investigated the classes of antibiotics, mostly dispensed without a prescription. In addition, we investigated the medical conditions to which antibiotics mainly were dispensed without a prescription. Pharmacists were asked to select a class of antibiotics from the list that they would easily dispense to a patient without a prescription suffering from suspected bacterial infection. They were also asked to select two medical conditions to which antibiotics were administered without a prescription based on their practice and experience.

The fifth section assessed pharmacists’ practice and determined the proportion of pharmacists who dispense antibiotics without a prescription in Tanzania. Pharmacists were questioned if they would dispense antibiotics to a person without a prescription. Pharmacists were also asked if they inform a patient about the side effects of antibiotics, drug-drug/food interactions and the importance of adherence to the dispensed course of antibiotics.

Ten questions covering legal requirements for dispensing antibiotics and knowledge of AMR were asked and scored, giving each correct response a score of ten (10) points. The items assessed if pharmacists were aware of different rules, regulations and laws that govern antibiotics’ dispensing. The questions also were designed to evaluate if pharmacists knew that AMR was the consequences of dispensing antibiotics without a prescription. Each correct answer was scored, after which an overall score was calculated per respondent. The maximum score was 100 points, while the minimum score was 0 points. A Likert scale was used to categorize knowledge whereby those who scored 80–100 points were labelled as excellent knowledge, 60–79 points were labelled very good knowledge, 40–59 points were labelled good knowledge. In contrast, those who scored 20–39 points were regarded to have poor knowledge and 0–19 points were labelled very poor knowledge.

Five questions were asked to assess pharmacists’ attitudes on dispensing antibiotics without a prescription. Each response was scored 10 for the positive attitude and 0 scores for a negative attitude. The overall score was recorded and those who scored 30–50 out of 50 were labelled to have a positive attitude. In contrast, those who scored less than 30 out of 50 were considered to have a negative attitude on dispensing antibiotics without a prescription. 

### 2.4. Statistical Analysis

Data were downloaded from REDCap, cleaned to remove duplicate records and incomplete responses and analyzed using Statistical Package for Social Sciences (SPSS); SPSS Statistics for Windows, version 20 (IBM Corp., Armonk, NY, USA). Descriptive statistical analysis was performed, and a Chi-square test for categorical data was used where a *p*-value of less than 0.05 was considered to be statistically significant.

### 2.5. Ethical Considerations

An ethical clearance to conduct this study was obtained from the Muhimbili University of Health and Allied Sciences Research and Publications Committee with reference number DA.25/111/01. After a thorough explanation of the study objectives, online consent was obtained before filling the questionnaire, and a pharmacist was free to reject participating in the study. The pharmacist’s name was not recorded (only codes were used) and all other personal information was handled with confidentiality throughout the research and during presentations. 

## 3. Results

### 3.1. Socio-Demographic Characteristics

A total of 1100 licensed pharmacists were contacted and requested to fill the questionnaire through an online invitation link. A total of 226 (20.5%) responses were recorded, of which 197 responses were complete and used for analysis. Out of these respondents, 140 (71.1%) were males and 57 (28.9%) were females (Table 1).

The highest education level for most of the respondents was a bachelor degree in pharmacy 153 (77.7%), while 41 (20.8%) had masters and only 3 (1.5%) respondents were PhD holders. Out of 197 pharmacists, 182 (92.4%) pharmacists completed their bachelor studies in Tanzania (Table 1). Others graduated from Angola, China, India, Mozambique, Poland, South Africa, and United Arab Emirates.

A total of 143 (72.6%) out of the 197 responding pharmacists acknowledged dispensing antibiotics without a prescription (Table 1). 

A total of 127 (64.5%) pharmacists worked in community pharmacies. Out of these respondents, who work in community pharmacies 13 (10.2%) were pharmacy owners, 84 (66.1%) were dispensing pharmacists and 30 (23.6%) were both owners and dispensing pharmacists (Figure 1). The majority, 68 (58.2%) of the respondents work in community pharmacies that are located in Dar es Salaam region and 11 (8.7%) in the Mwanza region of Tanzania (Figure 2). 

### 3.2. Pharmacists’ Knowledge

Pharmacists’ responses to questions assessing knowledge were as follows: 81.2% of respondents were aware that the dispensing of antibiotics without a prescription is illegal in Tanzania and 18.8% did not know that the practice is illegal. 103 (52.3%) pharmacists were not sure if there is a penalty to a person who dispenses without a prescription. More than 90.0% of pharmacists agreed that dispensing without a prescription contributes to the inappropriate use of antibiotics and the development of AMR. Of the 197 surveyed pharmacists, 77.7% had overall excellent knowledge, 15.7% very good knowledge and 3.6% poor knowledge (Figure 3).

### 3.3. Pharmacists’ Attitude towards Dispensing of Antibiotics without a Prescription

Concerning pharmacists’ attitudes, 41 (20.8%) had a negative attitude, while 156 (79.2%) pharmacists had a positive attitude on dispensing antibiotics without a prescription. A positive attitude was for those who thought dispensing without a prescription should be stopped as it has negative consequences to the patient and society in general, and other related questions (Table 2).

### 3.4. Pharmacists’ Practice towards Dispensing of Antibiotics without a Prescription

Out of 197 pharmacists, 143 (72.6%) agreed that they do dispense antibiotics to a patient without a prescription in their practice while 54 (27.6%) denied having dispensed antibiotics without a prescription. A large proportion of pharmacists who work in community pharmacies dispense without a prescription more than those who do not work in community pharmacies (*p* < 0.001) (Table 3).

The highest education level attained was found to influence the practice as 84.1% (37/44) pharmacists with high education (masters and PhD holder) admitted to dispensing without a prescription compared to 69.3% (106/153) bachelor of pharmacy degree holders who dispense without a prescription (chi-square *p*-value of 0.04). There was a strong association between pharmacists’ attitudes and dispensing without a prescription (*p* < 0.001, Table 4). Age (*p* = 0.308), work experience (*p* = 0.617) and country of graduation (*p* = 0.458) were not significantly associated with the dispensing of antibiotics without a prescription. In a subset of 127 pharmacists that work in community pharmacy 68 (81.0%) pharmacy owners were not significantly more likely to dispense antibiotics without a prescription compared to 36 (83.7%) of non-owners (*p*-value =0.451). The other 23 (18.1%) pharmacists who worked in community pharmacy did not agree to dispense antibiotics without a prescription.

Out of the 140 pharmacists who admitted to dispensing antibiotics without a prescription, 86 (61.4%) promised to stop this practice. They perceived this as a bad practice that contributes to the inappropriate use of antibiotics and can potentially influence the development and spread of AMR. While, 54 (38.6%) pharmacists denied stopping the practice in the future because they regarded the practice to be helpful to the patients, three respondents did not provide preference to stop the practice. Furthermore, pharmacists in the survey perceived that being knowledgeable can help a patient get the right antibiotics even without laboratory tests and a prescription. Some of the reasons which were given out by pharmacists for stopping and not stopping dispensing antibiotics without a prescription in the future are summarized in Table 5.

Other reasons attributed to pharmacists to dispense antibiotics without a prescription, included pharmacists 141 (71.6%), thinking that they are knowledgeable enough to dispense without a prescription, where pharmacists being in close relationship with a customer who needs antibiotics 87 (44.2%) and business nature of pharmacy that focuses on making more profit. Out of 197 pharmacists, 131 (66.5%) think that pharmacists will stop dispensing antibiotics without a prescription if there could be a regular inspection by regulatory authorities like the Pharmacy Council (PC) and the Tanzania Medicines and Medical Devices Authorities (TMDA). The majority of pharmacist claimed that they are knowledgeable enough to determine the right type of antibiotic for the condition presented by their patient. One pharmacist was quoted saying “Pharmacists are well equipped with knowledge on common diseases and for which tests are not necessary...pharmacists are drug experts hence can make the right choice of antibiotics if he or she knows the disease condition”. Four pharmacists who intend to continue the practice of dispensing without a prescription indicated that “Pharmacist Initiated Therapy” ideology is the reason for them to continue dispensing antibiotics without a prescription.

Patients based reasons included pushing patients to visit pharmacies without a prescription asking for antibiotics when they are sick were noted. The leading causes were the reluctance of spending many hours during the patients’ visit to the health facilities and wait for a consultation to complete 160 (81.2%). The next leading cause was the inability of patients to afford consultation fee and cost for a laboratory test.

### 3.5. Classes of Antibiotics that Were Mostly Dispensed without a Prescription

Pharmacists responded to the question of what antibiotics they would readily and easily dispense to a patient without a prescription suffering from suspected bacterial infection by selecting from the list. A respondent was allowed to choose more than one class of antibiotics that he/she will feel comfortable to dispense to a patient who has no prescription.

Penicillins 167 (84.5%) were the most dispensed class of antibiotics without a prescription, followed by macrolides 83 (42.1%) and fluoroquinolones 69 (35.0%) (Figure 4).

### 3.6. Medical Conditions and Infections for Which Antibiotics Are Commonly Administered without a Prescription

Pharmacists mentioned some infections to which antibiotics are usually dispensed without a prescription. The leading conditions were urinary tract infection 143 (72.6%), cough 112 (56.9%) and sexually transmitted diseases 85 (43.1%) (Figure 5).

## 4. Discussion

Inappropriate dispensing of antibiotics has been reported to be common in many LMICs [24]. A few surveys have been conducted in Tanzania to evaluate dispensing practices of antibiotics in community pharmacies, Accredited Drug Dispensing Outlets (ADDOs) and *Duka La Dawa Baridi (DLDB)* (translated as essential medicine drug outlets) and other shops. The surveys revealed antibiotics dispensed without a prescription making apparent breach of GDP. For instance, in a study conducted in Moshi municipality, in Kilimanjaro, Tanzania, 92.3% of pharmacy dispensers issued antibiotics without a prescription [17]. Although there are regulatory authorities like the TMDA and the PC who supervise ADDO and Pharmacy shops, there are still reports of dispensing mal-practice in ADDOs and DLDB’s. The cross-section survey in four different districts in Tanzania found that dispensers were issuing antibiotics without a prescription in ADDO shops and DLDBs [18]. Private rural drugstores do not adhere to GDP in dispensing antibiotics. Clients are issued antibiotics without a prescription after complaining to dispenser about their sickness like cough, flu and diarrhoea [19]. Tanzania regulations require a community pharmacy to be under the supervision of a licensed pharmacist. The pharmacist is responsible for making sure that the GDP protocols are adhered to during the dispensing of drugs. A previous study conducted in Tanzania revealed that both pharmacies and ADDOs dispensers issue antibiotics without a prescription.

To the best of our knowledge, this is the first study to evaluate pharmacists’ knowledge, attitude and practice regarding the dispensing of antibiotics without a prescription in Tanzania.

In this study, about 81.2% of pharmacists were aware that dispensing of antibiotics without a prescription is illegal in Tanzania contrary to a study conducted in Saudi Arabia, where only a quarter of pharmacists were aware [9]. Despite the majority of pharmacists having excellent knowledge about legal requirements to dispense antibiotics and consequences of the inappropriate use of antibiotics, 72.6% pharmacists agreed that they dispense without a prescription in their practice. These results were consistent with a study conducted in Syria, where 89.3% pharmacists admitted to dispensing antibiotics without a prescription [6]. In contrast, simulated research conducted in India and the Kilimanjaro region in Tanzania revealed that 66.7% and 90.0% pharmacies dispense antibiotics without a prescription, respectively [17,25].

Patients, pharmacists and the regulatory authorities’ practices may all contribute to this mal-practice. The tendency of patients to visit pharmacy without a legal prescription was claimed by pharmacists to be one of the reasons for them to dispense without a prescription to meet customer demands and expectations. Reluctance to spend many hours when they visit hospitals and inability to afford consultation fees and cost of laboratory tests were the leading factors that hinder the patient from obtaining a prescription. Similar reasons were revealed in a study conducted in Saudi Arabia and Khartoum Sudan [2,11]. Pharmacists seem to dispense antibiotics without a prescription to maximize sales and profit because they think the regulatory authorities are not stringent enough to prevent inappropriate dispensing of antibiotics. If a patient is denied antibiotics, she/he will obtain them in other pharmacies, as one pharmacist responded “if they don’t get it from my pharmacy they will get it from another pharmacy”, the same reasons were reported in other studies conducted in ADDOs in Tanzania to evaluate the reasons that motivate dispensers in ADDOs to dispense antibiotics without a prescription [26].

The personal conviction that pharmacists are knowledgeable enough about diseases and antibiotics was a reason for dispensing antibiotics without a prescription. Pharmacists who were masters and PhD holders were significantly more likely to dispense antibiotics without a prescription than the bachelor of pharmacy holders. This finding indicates that the higher the level of education, the more confident is the pharmacist to dispense without a prescription. Probably, at this education level, the pharmacist thinks that he/she knows enough to evaluate a patient sickness and treat accordingly, even giving antibiotics without a prescription. Some of the responses of pharmacists as to why they dispense without a prescription included the following quotes: “some conditions are so obvious that you know the patient sickness”, “pharmacists are drug experts hence can make a good choice of antibiotics if he or she is sure of the condition”, “I have the knowledge that will clinically guide me to dispense when there is a need and the infections is not that much serious”. Some pharmacists claimed that there is a Pharmacist Initiated Therapy (PIT) that allows them to dispense without a prescription. This was a misconception of the PIT, where only non-prescription drugs are allowed to be dispensed by a pharmacist without a prescription [27].

Antibiotics from penicillins, macrolides and fluoroquinolones classes were found to be the most dispensed antibiotics. These antibiotics were claimed to be used to treat medical conditions such as urinary tract infections, cough and sexually transmitted diseases. These findings are similar to other surveys conducted in Ethiopia and Egypt [28,29] and slightly different from those conducted in Saudi Arabia, where penicillins and cephalosporins were the most dispensed antibiotics without a prescription [2]. The observed difference may be due to the variability of the prevalence of diseases in the two countries as urinary tract infections and cough were the common medical conditions to which antibiotics were mostly dispensed without a prescription in Tanzania. In Saudi Arabia, cold and flu, rhinitis and toothache the most reported conditions [2]. The relatively low-cost penicillins may stimulate the patients to request that class of medicines more frequently compared to other antibiotics. Overuse of penicillins and macrolides increases bacterial resistance to these essential classes of antibiotics. The practice complicates future treatment choices for bacterial infections. The WHO has already reported the resistance of penicillins and fluoroquinolones in several countries [30]. The use of fluoroquinolones may be associated with adverse events compared to other antibiotics [31]. Besides, fluoroquinolones should be reserved for treatment of serious and potentially life-threatening infections and not for folks walking in to a community pharmacy. Therefore, fluoroquinolones should strictly be dispensed when prescribed by a qualified medical practitioner.

Preventing inappropriate use of antibiotics and the growth of AMR is the goal of any country’s health sector. AMR is caused by interconnected factors involving patients, health care providers and regulatory authorities. Coordinated interventions at both national and international levels are required in combating the AMR. Creating awareness about AMR in society will be a first step in promoting the rational use of antibiotics as proposed in a national action plan on antibiotic resistance [32]. From the findings of this study, the government should consider decreasing the consultation fee and laboratory test costs as these were the main hindrance for patients to get a prescription. Strict regulatory authorities and regular inspections are the core for tackling antibiotics’ dispensing without a prescription in pharmacies. In countries where there are strict functioning regulatory authorities, the malpractices in pharmacies have been observed to decline [33,34].

In this study, there was a significant more proportion pharmacist with a bad attitude (97.6%) compared to 66.0% with a good attitude who are likely to dispense antibiotics without a prescription. Recently we assessed the implementation of Antimicrobial Resistance Surveillance and Antimicrobial Stewardship Programs in Tanzania health facilities one year after the launch of the National Action plan in Tanzania [8]. We noted that even though all 39 (100%) respondents were aware of the presence of AMR in Tanzania, only 26 (66.7%) were aware of the presence of the Tanzanian National Action Plan (NAP) for AMR. In that survey, the majority of respondents were clinical pharmacists 30 (76.9%), indicating that pharmacists are key players in the implementation of Antimicrobial Resistance Surveillance and Antimicrobial Stewardship Programs in Tanzania and beyond. Therefore they should act as pioneers in the fight against AMR. This will, however, require more training of the pharmacists to keep them updated of the NAP and change their attitude.

There are some limitations to this study, pharmacists were invited to fill the online questionnaire via an invitation link and there was no physical contact between the respondent and the investigator. However, we are certainly sure that pharmacists are the ones who filled the questionnaire because the link was shared only to pharmacists after obtaining their contacts from the Pharmacy Council, the authority responsible for the control, regulation and management of the pharmacy profession and practice in Tanzania. An additional limitation is a fact that all licensed pharmacists were included in the study regardless of working in the community pharmacies or not. This inclusion intended to investigate the attitude and remarks of all pharmacists regarding the dispensing of antibiotics without a prescription. The majority of respondents were from Dar es Salaam, a city with the highest population in Tanzania and few respondents from other regions. More than 50% of community pharmacies are located in Dar es Salaam and the rest in other upcountry regions, making the findings of this study to be valid to represent the whole country.

## 5. Conclusions

The Tanzanian pharmacists seem to be knowledgeable of the legal requirements to dispense antibiotics, the antimicrobial resistance threat and its consequences to both patient and the public. However, most of these pharmacists admitted to dispensing antibiotics without a prescription and claimed that the practice is common in numerous community pharmacies in Tanzania. The business nature of pharmacy, a failure of the patient to get a prescription and the lack of rigorous regulatory authorities’ actions were cited as the main reasons for this violation. The antibiotics frequently reported to be dispensed without a prescription were penicillins, macrolides and fluoroquinolones. These antibiotics were used to treat medical conditions such as urinary tract infections, cough and sexually transmitted diseases.

There is an obvious need for educating the community regarding antimicrobial resistance and the importance of visiting the nearby health center for consultation and laboratory tests before obtaining medications. Community awareness will help in tackling the inappropriate use of antibiotics, emergence and spread of antimicrobial resistance. On the one hand, the government should consider reducing the consultation fee and laboratory test costs in public health centers. These fees are a considerable barrier for the majority of patients to afford. On the other hand, the regulatory authorities should make regular inspections to pharmacies to detect any prevailing malpractice and insist that pharmacists stop dispensing antibiotics without a prescription. In case a medicine dispenser is caught intentionally issuing prescription-only medicines without a prescription, the person should be penalized according to the country’s current pharmacy profession laws.

## Figures and Tables

**Figure 1 pharmacy-08-00238-f001:**
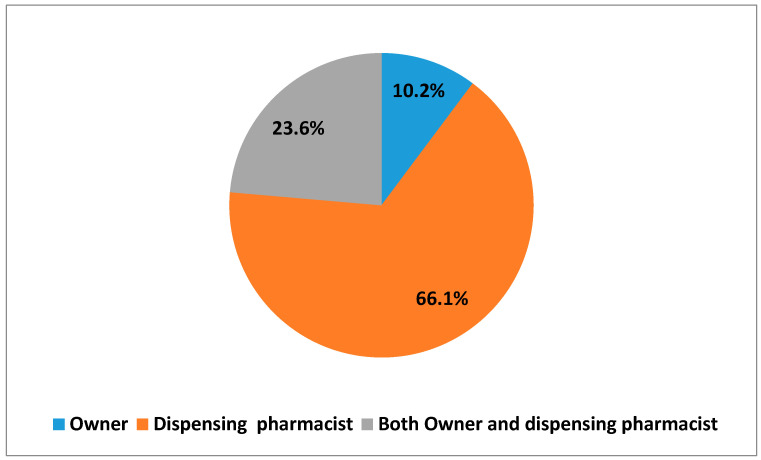
Job-status in community pharmacy (*n* = 127)**.**

**Figure 2 pharmacy-08-00238-f002:**
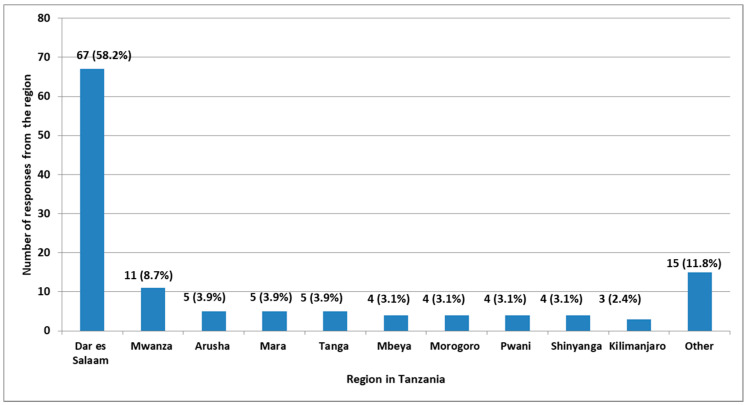
Region of the practice of community pharmacists in Tanzania (*n* = 127). Other regions were Lindi, Ruvuma, Singida, Tabora, Dodoma, Iringa, Kagera, Kigoma, Mtwara, Songwe, and Mjini Magharibi which each with less than three responses.

**Figure 3 pharmacy-08-00238-f003:**
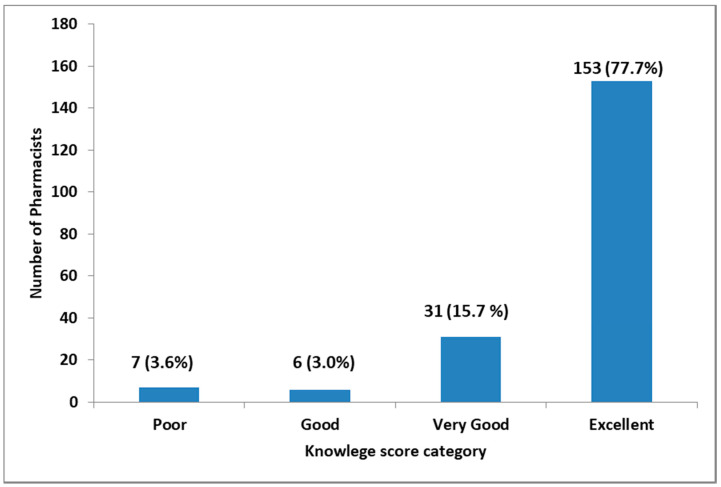
The overall pharmacist knowledge of legal needs to dispense antibiotics and AMR. The Likert scale was used to categorize pharmacists based on their scores on the ten questions to assess pharmacist knowledge (*n* = 197).

**Figure 4 pharmacy-08-00238-f004:**
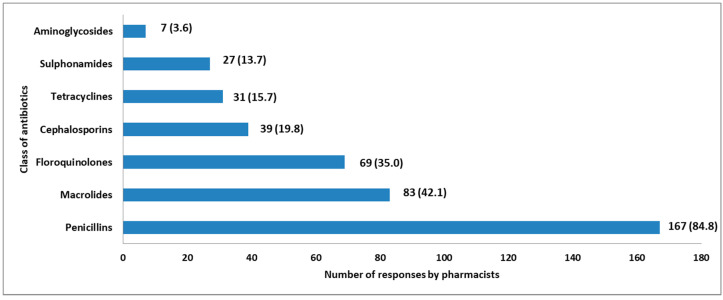
Class of antibiotics mostly dispensed without a prescription.

**Figure 5 pharmacy-08-00238-f005:**
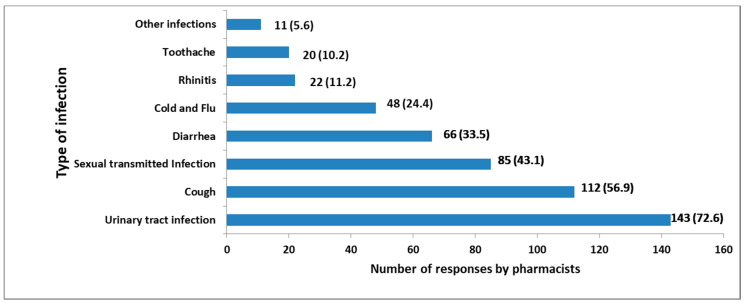
Medical conditions and infections to which antibiotics are commonly dispensed without a prescription as suggested by pharmacists.

**Table 1 pharmacy-08-00238-t001:** Social-demographic data of Tanzanian pharmacists responding to the questionnaire.

Parameter	Frequency (%)
**Age, years (*n* = 197)**	
21–30	94 (47.7)
31–40	70 (35.5)
>40	33 (16.8)
**Experience, years (*n* = 197)**	
1–5	125 (63.5)
>5	72 (36.5)
**Sex (*n* = 197)**	
Male	140 (71.1)
Female	57 (28.9)
**Education level (*n* = 197)**	
Bachelor	153 (77.7)
Masters	41 (20.8)
PhD	3 (1.5)
**Country of graduation (*n* = 197)**	
Tanzania	182 (92.4)
India	8 (4.1)
Other	7 (3.5)
**Dispense antibiotics without a prescription? (*n* = 197)**	
Yes	143 (72.6)
No	54 (27.4)
**Work in any community Pharmacy? (*n* = 197)**	
Yes	127 (64.5)
No	70 (35.5)

**Table 2 pharmacy-08-00238-t002:** Pharmacists’ attitudes towards dispensing antibiotics without a prescription (*n* = 197).

Question Responses	Yes	No
Do you think there is any problem if you dispense antibiotics without a prescription?	186 (94.4%)	11 (5.6%)
Pharmacist should stop dispensing without a prescription	156 (79.2%)	41 (20.8%)
I encourage the patient to consult a physician and get a prescription before visiting the pharmacy	111 (56.3%)	86 (43.6%)
Pharmacists are knowledgeable enough to dispense without a prescription after critical evaluation of patient sickness	141 (71.6%)	56 (28.4%)
Do you dispense antibiotics without a prescription?	143 (72.6%)	54 (27.6%)

**Table 3 pharmacy-08-00238-t003:** Comparison of dispensing without a prescription practice among pharmacists who work predominantly as community pharmacist compared other pharmacists (*n* = 197).

	Work as Community Pharmacist?	No	Yes	Total	*p*-Value
Dispense without a prescription?	Yes	39 (55.7%)	104 (81.9%)	143 (72.6%)	
	No	31 (44.3%)	23 (18.1%)	54 (27.4%)	
	Total	70 (100%)	127 (100%)	197 (100%)	<0.0001

**Table 4 pharmacy-08-00238-t004:** Comparison of dispensing without a prescription practice among pharmacists with bad attitude versus those with a good attitude (*n* = 197).

	Attitude	Bad	Good	Total	*p*-Value
Dispense without prescription?	Yes	40 (97.6%)	103 (66.0%)	143 (72.6%)	
	No	1 (2.4%)	53 (34.0%)	54 (27.4%)	
Total		41 (100%)	156 (100%)	197 (100%)	<0.0001

**Table 5 pharmacy-08-00238-t005:** Pharmacists’ reasons for stopping or not stopping dispensing antibiotics without a prescription.

Reasons for Stopping Dispensing Antibiotics without Prescription (*n* = 83) *	Frequency (%)
To prevent the development and spread of AMR	46 (55.4)
To promote rational use of medicines	13 (15.6)
There are stringent regulatory authorities	11 (13.2)
The practice is illegal	9 (10.8)
There will be adequate health facilities in the future	4 (5.0)
**Reasons for not stopping the practice in the future (*n* = 57) ***	
Pharmacists are knowledgeable enough to dispense without a prescription	29 (50.9)
Profitability nature of pharmacies business	12 (21.1)
Not enough health facilities for patients to obtain a prescription	9 (15.8)
No stringent regulatory authorities	7 (12.2)

Key: * indicate that responses were only given by pharmacists who indicated to dispense antibiotics without a prescription (*n* = 143) and plan to stop (*n* = 86) or not stop (*n* = 57) the practice. One person did not provide reasons for stopping the practice.

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
