# Peer review of "Pharmacists’ Knowledge, Attitude and Practice Regarding the Dispensing of Antibiotics without Prescription in Tanzania: An Explorative Cross-Sectional Study"

_pharmacy, 2020, doi:10.3390/pharmacy8040238_

Round 1

Reviewer 1 Report

Thank you for the opportunity to review this paper. This was a timely read, especially given the current climate and global push towards combating AMR. The authors have made a case for the study from their introduction, which clearly describes why the study is needed within the specified context. 

There were no issues of significant concerns with respect to the research methodology other than a few points below:

  • Please include a statement about the validation process taken for the survey tool. Also, it is unclear if this was a completely new tool developed for the study, or an adapted tool.
  • Either way, the authors need to make a statement about its validation.

In the result section:

  • There seem to be discrepancies in the number of participants reported. Line 159, 161, 196 and Table 1 all carry different total sample sizes. This need reviewing and correcting by the authors.
  • Line 209, do you mean that they indicated willingness to stop the practice? At present, it is unclear what the authors mean by “promised to stop”.
  • The way the results are reported is a bit unclear. How different were the responses between the community and non-community pharmacists? It might be useful to add a few more comparisons here about the responses between the two distinct cohorts represented.

Other than these, I think you paper is well presented.

Best wishes

Author Response

Dear Reviewer, Dear Editor

We thank you for the comprehensive comments that you made to our manuscript, which has contributed to the substantial improvement of the manuscript.

We hereby attach the revised manuscript along with responses to your comments.

Sincerely

Reviewer 2 Report

Sentence structure and grammar could be improved in places. The methodology and results sections need re-writing and editing in places. There appears to be some inconsistencies in the number of respondents in the results section i.e. did 213, 212, 211 or 206 pharmacists complete the survey. I have made comments on the attached version of the submission for you to consider.

Author Response

(The authors gave the same response as above.)

Reviewer 3 Report

I have attached the file with minor improvement suggestions- I don't see any where if ethics applications was applied for and obtained. Also was confidentiality of participants observed. other than that some sentences appear to be confusing, mainly because these are too long. 

Author Response

(The authors gave the same response as above.)

Round 2

Reviewer 2 Report

There are still a significant number of English grammar and sentence structure issues in the transcript. Some are highlighted in the attached document. 

Author Response

Dear Reviewer, Dear Editor

We thankyou for the second round of revisions which led to the improvement of our manuscript.

The responses to the comments are attached in the document

Thank you for considering our manuscript
